# Real-World Impact of SABR on Stage I Non-Small-Cell Lung Cancer Outcomes at a Scottish Cancer Centre

**DOI:** 10.3390/cancers15051431

**Published:** 2023-02-23

**Authors:** Mark Stares, Georgina Lewis, Maheva Vallet, Angus Killean, Giovanni Tramonti, Ailsa Patrizio, Melanie Mackean, Stephen Harrow, Colin Barrie, Kirsty MacLennan, Sorcha Campbell, Tamasin Evans, Aisha Tufail, Peter Hall, Iain Phillips

**Affiliations:** 1Edinburgh Cancer Centre, NHS Lothian, Western General Hospital, Crewe Road South, Edinburgh EH4 2XU, UK; 2Cancer Research UK Edinburgh Centre, Institute of Genetics and Cancer, Western General Hospital, University of Edinburgh, Crewe Road South, Edinburgh EH4 2XR, UK

**Keywords:** non small-cell lung cancer, stage I, stereotactic ablative radiotherapy, real-world clinical data

## Abstract

**Simple Summary:**

Surgery or conventional radical radiotherapy are established curative treatment options for patients with localised, stage I non-small-cell lung cancer (NSCLC). Another option, stereotactic ablative body radiotherapy (SABR), has emerged over the last decade. We sought to understand the impact of the introduction of SABR at our institute by investigating outcomes in 1143 patients diagnosed with stage I NSCLC between 2012–2019. We find that clinical factors such as age, fitness and the presence of other significant health problems (i.e., comorbidities) correlated with treatment choice. For example, younger, fitter and less comorbid patients were more likely to be treated with surgery. Similar to other studies in this area, we find that the introduction of SABR improved survival outcomes of patients with stage I NSCLC. The greatest improvement was seen in patients treated with surgery. We suggest this is because the availability of SABR as a safe and effective alternative treatment improved the selection of patients for surgical treatment. These findings are important as they may help clinicians and patients chose the most appropriate treatment option.

**Abstract:**

Introduction: Stereotactic ablative body radiotherapy (SABR) offers patients with stage I non-small-cell lung cancer (NSCLC) a safe, effective radical therapy option. The impact of introducing SABR at a Scottish regional cancer centre was studied. Methods: The Edinburgh Cancer Centre Lung Cancer Database was assessed. Treatment patterns and outcomes were compared across treatment groups (no radical therapy (NRT), conventional radical radiotherapy (CRRT), SABR and surgery) and across three time periods reflecting the availability of SABR (A, January 2012/2013 (pre-SABR); B, 2014/2016 (introduction of SABR); C, 2017/2019, (SABR established)). Results: 1143 patients with stage I NSCLC were identified. Treatment was NRT in 361 (32%), CRRT in 182 (16%), SABR in 132 (12%) and surgery in 468 (41%) patients. Age, performance status, and comorbidities correlated with treatment choice. The median survival increased from 32.5 months in time period A to 38.8 months in period B to 48.8 months in time period C. The greatest improvement in survival was seen in patients treated with surgery between time periods A and C (HR 0.69 (95% CI 0.56–0.86), *p* < 0.001). The proportion of patients receiving a radical therapy rose between time periods A and C in younger (age ≤ 65, 65–74 and 75–84 years), fitter (PS 0 and 1), and less comorbid patients (CCI 0 and 1–2), but fell in other patient groups. Conclusions: The introduction and establishment of SABR for stage I NSCLC has improved survival outcomes in Southeast Scotland. Increasing SABR utilisation appears to have enhanced the selection of surgical patients and increased the proportion of patients receiving a radical therapy.

## 1. Introduction

Lung cancer is the leading cause of cancer death in Scotland, accounting for one in five cancer deaths [1]. Non-small-cell lung cancer (NSCLC) represents approximately 85% of all cases [2]. In Scotland, approximately 20% of patients present with stage I disease, typified by small (<4 cm) localised disease without spread to lymph nodes or distant organs [3]. Surgical resection, involving lobectomy with mediastinal lymph node dissection or sampling, has been the curative treatment of choice for stage I NSCLC. However, many patients with lung cancer are burdened by multiple co-morbidities, including chronic obstructive pulmonary disease (COPD) or cardiovascular disease, which make them less suitable for surgery [4,5].

Non-surgical treatment options such as conventional fractionated radical radiotherapy (CRRT) may also be used with radical intent. However, historically, outcomes are poorer than those achieved by surgery [6,7]. More recently, stereotactic ablative body radiotherapy (SABR) has become the treatment of choice in patients who are unfit for surgery or decline resection [8]. SABR is a well-tolerated and effective treatment in these patients [9,10,11]. Registry data suggest SABR improves survival when compared to best supportive care [12]. When compared to standard CRRT, SABR is more convenient for patients, has no minimum threshold for respiratory function, fewer side effects, a higher local control rate and is likely to have a survival benefit [13,14,15,16]. Unfortunately, randomised controlled trials of SABR vs. surgery have struggled to recruit, largely due to patient preference for radiotherapy over surgery or vice versa [17,18]. However, in younger, fitter patients, surgical resection would be considered the standard of care [19,20]. For patients who are potentially operable, SABR and surgery outcomes appear to be similar in the limited trial data available [18,21]. This suggests SABR is a reasonable alternative to surgery in those who decline an operation, or in those who have a higher risk of surgical complications. A key benefit of SABR is that it increases the pool of patients who could receive an effective radical treatment [13,22]. In a previous observational cohort study, the use of SABR increased the proportion of older patients, at the highest risk of surgical complications, who received a radical treatment [23,24]. Consequently, the average survival of the whole cohort increased.

The aim of this study was to understand the impact of SABR on outcomes for stage I NSCLC at a regional cancer centre in the United Kingdom (UK). We demonstrate the positive effect of the introduction of SABR as a treatment option for stage I NSCLC in a real-world setting. We present novel data demonstrating the impact of clinical factors on treatment selection and outcomes. Interestingly, we find that the availability of an alternative low-toxicity treatment to surgery appears to affect the selection of surgical patients, leading to improved surgical outcomes.

## 2. Methods

All NHS Lothian patients discussed in the Southeast Scotland Cancer Network (SCAN) lung-cancer multidisciplinary meeting between January 2012 and December 2019, diagnosed clinically with a stage I NSCLC, were identified [25]. Patients with multiple synchronous or metachronous primary lung cancers were excluded. Patients upstaged at surgery were included in analyses based on an intention to treat as stage I NSCLC. Data were extracted from the Edinburgh Cancer Centre Lung Cancer Database, containing detailed clinical information for all patients with lung cancer across SCAN since 2012.

Patient characteristics, including age, Eastern Cooperative Group Performance Status (PS) and Charlson Comorbidity Index (CCI) at the time of diagnosis of stage I NSCLC and treatment modality were recorded [26]. CCI was calculated using hospital admission data obtained from the Scottish Morbidity Records dataset [27]. CCI was grouped by no comorbidity (CCI 0), mild/moderate comorbidity (CCI 1–2) or significant comorbidity (CCI ≥ 3).

Radical radiotherapy treatment status was defined as: CRRT—55 Gy in 20 fractions as fractionated dose; SABR—54 Gy in 3 fractions, 55 GY in 5 fractions or 60 Gy in 8 fractions. This is in keeping with previously reports [14].

Three distinct time periods were studied reflecting the availability of treatment options within SCAN: A—January 2012–December 2013 (pre-SABR); B—January 2014–December 2016 (introduction of SABR); C—January 2017–December 2019, (SABR established).

The overall survival, defined as the number of months from the date of diagnosis of stage I NSCLC and death, or censorship if still alive at follow-up (1 November 2021), was calculated. Survival curves were plotted using Kaplan Meier methods, and the log rank test applied. Survival analysis was carried out using Cox’s proportional-hazards model, and hazard ratios were calculated. Differences in treatment groups and time periods were compared using *t*-tests for continuous variables and chi-square tests for categorical variables as appropriate. A *p*-value < 0.05 was considered significant throughout. All analyses were performed in SPSS version 27.0 (SPSS Inc).

## 3. Results

Patient Characteristics: 1143 patients meeting the inclusion criteria were identified. Patient characteristics were in keeping with reported real-world populations of stage I NSCLC (Table 1). A total of 41 (9%) patients treated with surgery were upstaged. Analyses of all patients diagnosed with NSCLC within NHS Lothian during the study time periods demonstrated no evidence of stage migration (Appendix A). The median age was 74 (interquartile range (IQR) 68–81) and 55% were female.

Median OS was 41.6 (interquartile range (IQR) 15.4–95.8) months. A total of 407 (36%) patients were censored in whom minimum and median follow-up was 26.9 and 58.4 months, respectively. Age (≤65, 65–74, 75–84, ≥85 years old), PS (0, 1, 2, 3+) and CCI (0, 1–2, 3) were independently associated with survival (Appendix A) (each log-rank *p* < 0.001).

Surgery was the most frequently employed treatment modality (41%). Age, PS and comorbidities were important factors for treatment choice (Appendix A). Patients treated with surgery were younger (median age 70 (IQR 63–75) vs. 78 (IQR 72–84), *p* < 0.001), of better PS (PS0/1 86% vs. 50%, *p* < 0.001) and less comorbid (CCI 0 54% vs. 45%, *p* < 0.001) than all other patients. A total of 82% of patients aged ≤65 and PS0 were treated surgically, whilst 74% of those aged ≥85 and PS2+ received no radical treatment (Appendix A).

Outcomes by Treatment Modality: As expected, patients with no radical treatment had the poorest survival (13.5 (IQR 5.3–30.3)) (Figure 1, Appendix A). Outcomes for patients treated with surgery (92.3 (IQR 40.6—not reached)) were more favourable than those treated with SABR (65.3 (IQR 29.1–85.3), which were more favourable than those treated with CRRT (37.1 (IQR 18.5–59.6)) (*p* < 0.001 and *p* < 0.001, respectively).

Outcomes by Time Period: Survival estimates by time period for all patients, and for each treatment subgroup, are shown in Figure 2 (Appendix A). Patients in time period C had more favourable survival than those in time period A (HR 0.85 (95% confidence interval (CI) 0.77–0.94)), with median survival improving from 32.5 (IQR 13.0–74.8) months to 48.8 (15.3–95.8) months (*p* = 0.006) (Appendix A). The greatest improvement in survival was seen in patients treated with surgery between time periods A and C (HR 0.69 (95% CI 0.56–0.86), *p* < 0.001). The survival of patients treated with any radical radiotherapy (i.e., CRRT or SABR) improved between time periods A and B (HR0.70 (95% CI 0.49–0.99), *p* = 0.045) and between time periods A and C (HR0.75 (95% CI 0.61–0.91), *p* = 0.004).

Patient selection with increasing availability of SABR: Changes in treatment patterns were observed across time periods (Figure 3). The proportion of patients who received no radical therapy fell from 33% to 30% amongst all patients, and from 52% to 46% in the elderly (≥75 years old) population (Appendix A). SABR use rose from 11% to 18% between time periods B and C in all patients, offset by stepwise reductions in the use of CRRT and surgery.

Changes in treatment patterns were observed between time periods by age group, PS and CCI (Figure 4). The proportion of patients receiving a radical therapy rose between time periods A and C in younger (age ≤ 65, 65–74 and 75–84 years), fitter (PS 0 and 1) and less comorbid patients (CCI 0 and 1–2). In each of these patient cohorts, the use of CRRT and surgery fell between time periods A and C, with SABR increasingly utilised between time periods B and C. In older (aged ≥ 85 years), less fit (PS 2) and more comorbid patients (CCI ≥ 3) fewer patients received a radical therapy in time period C than time period A. In each of these patient groups, the use of CRRT and surgery also fell between time periods A and C.

There were no statistically significant differences in patient characteristics for each treatment group between time periods (*p* > 0.05).

## 4. Discussion

Our real-world data demonstrate an increase in the proportion of patients with stage I NSCLC receiving a radical therapy between 2012 and 2019. The median overall survival of the study population increased by 16.3 months between time periods A and C, with the most significant improvement was seen in patients undergoing surgical management of their cancer. These changes correlated with the introduction and establishment of SABR as a standard treatment option at the Edinburgh Cancer Centre. This is the first time this has been demonstrated in a UK population. Our findings largely reflect those previously demonstrated in a Dutch population-based study, which found that the introduction of SABR correlated with a decline in the number of untreated elderly patients with stage I NSCLC, corresponding to an 8-month improvement in median overall survival [23].

A key clinical challenge is to improve radical treatment rates for patients with stage I NSCLC. In a 2015/16 Cancer Registry analysis, rates of no radical therapy were 26% in England, 13% in Norway and 9% in the Netherlands [28]. Significantly, in that study, only 8% of patients in England were treated with SABR, compared to 26% in Norway and 27% in the Netherlands, reflecting the slower establishment of SABR in the UK. Our rates of SABR remain lower than this (18%), despite SABR now being an established treatment. Previous studies examining the impact of SABR on the management of stage I NSCLC have lacked recognised clinical prognostic factors such as PS and detailed comorbidity data [23,24,28]. We find that age, PS and comorbidity burden, as measured by the CCI, are associated with overall survival outcomes in this population. We present novel data demonstrating that treatment patterns strongly correlated with these factors. For example, surgical rates were lower with increasing age, whilst any radical radiotherapy (CRRT or SABR) use became more frequent. Significantly, the commonest treatment for patients ≥75 years in our study was no treatment (49%), whereas 85% of those <75 years received a radical therapy. We add to this by demonstrating that patients with poor PS or significant comorbidities are also less likely to be treated radically. In particular, these patients are less frequently treated with surgery. Pre-existing respiratory comorbidities, such as COPD, may increase the risk of post-operative complications, limit the extent of lung that can be safely removed and are associated with poorer outcomes in stage I NSCLC [10].

We also note that between time periods A and C, rates of radical therapy increased by only 3% in the overall population and 6% in patients ≥75 years old. This is lower than that seen in a previous real-world observational study [24]. Given that patients in the NRT cohort were older, less fit and more comorbid, we suspect that many had incidental lesions identified but were not fit for further investigation and management. Our institution has no thresholds for minimum lung function for SABR and, broadly, if a patient tolerates PET-CT they are likely to tolerate the delivery of SABR. Indeed, only 2% of patients in the NRT received any direct cancer palliative therapy, including high-dose palliative radiotherapy. This suggests that, in addition to the availability of new treatments, strategies to improve patient fitness or the early detection of cancer are needed to improve radical treatment rates.

In our clinical practice, surgery remains the treatment of choice for patients with stage I NSCLC. That patients treated with surgery in our cohort had significantly better survival than those treated with any radical radiotherapy likely reflects differences in treatment selection. SABR offers an alternative treatment option for patients with high surgical risk and technically and medically inoperable disease. Specifically, it is associated with lower 30-day mortality than surgery in patients with severe COPD, but offers similar survival benefit [10]. It is also proven to be a better treatment than CRRT (the only other pre-existing non-surgical radical treatment option) with fewer side effects, higher rates of local control and a likely survival benefit [16,17,18,19]. It appears to be well tolerated in older, frailer patients [22,29,30]. It is, therefore, not surprising to find that SABR use increased at the expense of CRRT and surgery in these patient groups. Indeed, rates of any radical therapy fell in these patient groups, but increased in younger, fitter or less comorbid patient groups. Although this may reflect a better selection of patients for radical therapy, which may have contributed to better survival between time periods A and C, we suggest these changes were driven by the introduction of SABR for the treatment of stage I NSCLC. For example, we demonstrate that the introduction of SABR correlated most strongly with a survival improvement for patients treated with surgery. A potential confounder to these findings is the improvement in surgical techniques and perioperative care during the study time periods. However, the most significant reductions in surgical rates between time periods A and C were seen in patients aged ≥ 85 years (5% vs. 0%), PS 2 (28% vs. 18%) and CCI ≥ 3 (24% vs. 7%). This likely reflects the availability of an additional efficacious treatment option and highlights that an important real-world impact of SABR has been to facilitate better selection of patients for surgery. This effect of improving outcomes by the migration of the poorer outcome patients into a different group is recognised in the staging of cancer and is known as the Will Rogers effect, first described in 1985 [31]. This has not previously been described for the surgical treatment of NSCLC. Although survival improved between time periods A and C for patients treated with any radical radiotherapy, there was no significant change for patients treated with CRRT, suggesting this improvement was driven by treatment with SABR.

CRRT and surgical rates fell in all other patient characteristic subgroups between time periods A and C, with SABR utilised in each. This suggests that SABR has an important role to play in younger, fitter patients too. Use of SABR instead of CRRT in these subgroups may reflect the availability of a better treatment option than CRRT, particularly where surgery is not possible for technical reasons. It is also recognised that patients, when offered the choice, frequently opt for SABR over surgery [17,18]. Significantly, amongst treated patients, SABR was the most frequently applied radical therapy in those with mild functional limitations (i.e., PS1) and mild/moderate comorbidities (CCI 1–2) (45% and 40%, respectively), where the clinical assessment of suitability for surgery is less clearcut between operable and inoperable. There is longstanding debate around the role of SABR in potentially operable patients, particularly as many of these patients are older or more comorbid [32,33,34]. The positive real-world effects of the introduction of SABR identified by our study provides some evidence to fill the void left by the lacking clinical trials data in these patients. Our findings may become more important if computed tomography-based lung-cancer screening is introduced into routine clinical practice. The NELSON trial showed an overall survival benefit in the screened population compared to a control group (HR0.76 (95% CI 0.61–0.94), *p* = 0.01) [35]. Significantly, there was a large increase in the proportion of patients presenting with stage I NSCLC (58.6% vs. 14.2%, respectively), suggesting the absolute number of patients with stage I NSCLC being considered for radical therapy may well increase if screening is introduced. A better understanding of factors important for treatment selection and outcomes, as explored in this study, will aid service provision.

Several limitations for this study are acknowledged. As a single-centre study, it benefits from standardised, comprehensive data collection of all patients with NSCLC, although some information on performance status and comorbidities was not available. The experience of the SCAN lung-cancer multidisciplinary team may have given rise to confounders in patient clinical selection for specific therapies. However, we observe differences in treatment selection through time, suggesting these are not inherent. Like other studies in this area, we have included patients without pathological confirmation, potentially including cases of benign disease, or isolated pulmonary metastases from another cancer. Our clinical practice, however, routinely includes the use of the Herder score and patients are staged with PET-CT imaging in line with UK guidelines [19]. In a previous study, 46% of all English patients with stage I NSCLC were treated with CRRT without histology, compared to 52% in our study [28].

## 5. Conclusions

This comprehensive study demonstrates how the introduction and establishment of SABR for stage I NSCLC has improved treatment rates and survival outcomes of patients in Southeast Scotland. We highlight recognised clinical prognostic factors that are key for patient treatment selection, which are absent from other similar studies. It is of particular significance that increasing SABR provision appears to have enhanced the selection of surgical patients, amongst whom survival outcomes are most improved. These findings support those of previous studies, suggesting the effects may be seen more broadly. SABR is now routinely available elsewhere, including at all five Scottish radiotherapy centres.

## Figures and Tables

**Figure 1 cancers-15-01431-f001:**
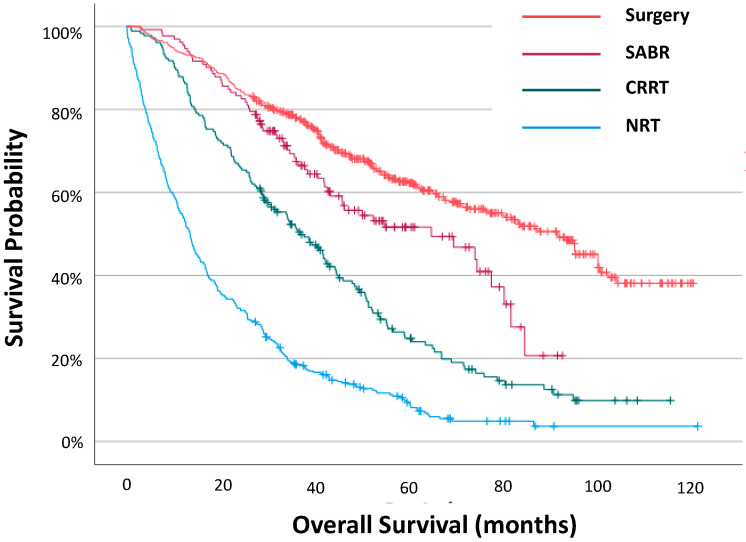
Overall survival for all patients with stage I NSCLC by treatment modality.

**Figure 2 cancers-15-01431-f002:**
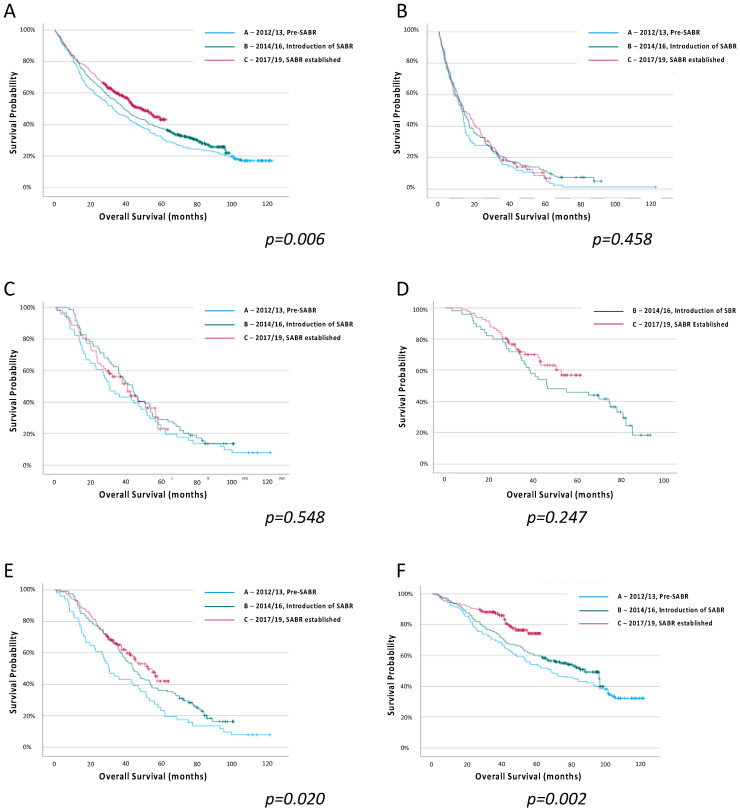
Overall survival estimates for all patients with stage I NSCLC, and for each treatment group, by treatment time period, (**A**) All patients, (**B**) No radical therapy, (**C**) Conventional radical radiotherapy, (**D**) SABR, (**E**) Any radical radiotherapy (CRRT or SABR), (**F**) Surgery. Log-rank regression.

**Figure 3 cancers-15-01431-f003:**
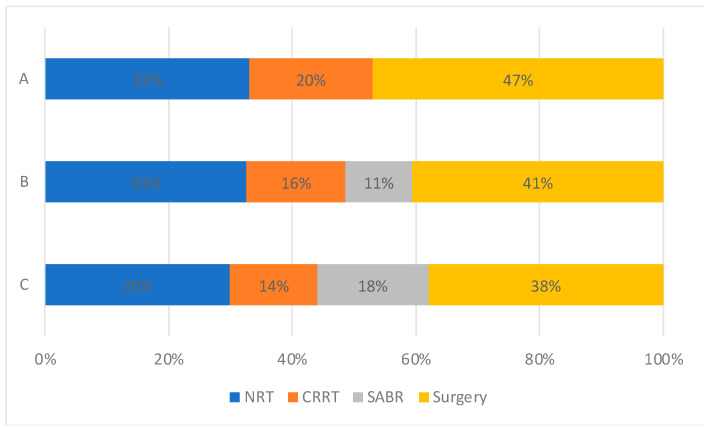
Treatment utilisation for all patients with stage I NSCLC by time period. (A: 2012–2103, Pre-SABR, B: 2104–2106, Introduction of SABR, C: 2017–2019, SABR Established).

**Figure 4 cancers-15-01431-f004:**
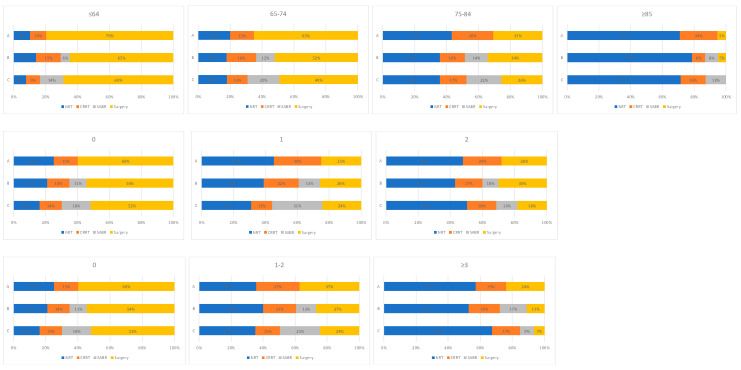
Treatment utilisation for all patients with stage I NSCLC by time period and A: age, B: performance status, C: Charlson comorbidity index subgroups. (A: 2012–2103, Pre-SABR, B: 2104–2106, Introduction of SABR, C: 2017–2019, SABR Established).

**Table 1 cancers-15-01431-t001:** Patient characteristics of patients with stage I NSCLC. (NR—not reached).

Patient Characteristics		All	No Radical Treatment	Radical Radiotherapy	Stereotactic Ablative Body Radiotherapy	Surgery
		*n* = 1143	*n* = 361	*n* = 182	*n* = 132	*n* = 468
Age	≤64	200 (17)	21 (6)	23 (13)	17 (13)	139 (30)
	65–74	372 (33)	66 (18)	60 (33)	44 (33)	202 (43)
	75–84	411 (36)	154 (43)	75 (41)	58 (44)	124 (26)
	≥85	160 (14)	120 (33)	24 (13)	13 (10)	3 (1)
	Median (IQR)	74 (68–81)	82 (75–87)	76 (70–81)	75 (69–81)	70 (63–75)
Sex	Female	628 (55)	200 (55)	95 (52)	75 (57)	258 (55)
	Male	515 (45)	161 (45)	87 (48)	57 (43)	210 (45)
ECOG Performance Status	0	244 (21)	51 (14)	21 (12)	22 (17)	150 (32)
	1	435 (38)	87 (24)	95 (52)	61 (46)	192 (41)
	2	225 (20)	66 (18)	54 (30)	42 (32)	63 (14)
	3+	76 (7)	76 (21)	0 (0)	0 (0)	0 (0)
	Unknown	163 (14)	81 (22)	12 (7)	7 (5)	63 (14)
Charlson Comorbidity Index	0	564 (49)	113 (31)	80 (44)	64 (48)	307 (66)
	1–2	301 (26)	114 (32)	60 (33)	44 (33)	83 (18)
	≥3	103 (9)	62 (17)	19 (10)	10 (8)	12 (3)
	Unknown	175 (15)	72 (20)	23 (13)	14 (11)	66 (14)
Pathological Confirmation	Yes	660 (58)	85 (23)	86 (47)	21 (16)	468 (100)
	No	483 (42)	276 (77)	96 (53)	111 (84)	0 (0)
T-stage	IA	783 (69)	247 (68)	99 (54)	112 (85)	325 (69)
	IB	360 (31)	114 (32)	83 (46)	20 (15)	143 (31)
Overall survival	Median (IQR)	41.6 (15.4–95.8)	13.5 (5.3–30.3)	37.1 (18.5–59.6)	65.3 (29.1–85.3)	92.3 (40.6-NR)
2-year survival	*n* (%)	744 (65)	116 (32)	121 (66)	110 (83)	397 (85)
Censored	*n* (%)	407 (36)	30 (8)	40 (22)	69 (52)	268 (57)
Period of Diagnosis	A (2012–2013)	252 (22)	83 (23)	51 (28)	0 (0)	118 (25)
	B (2014–2016)	443 (39)	144 (40)	69 (38)	50 (38)	180 (39)
	C (2017–2019)	448 (39)	134 (37)	62 (34)	82 (62)	170 (36)

## Data Availability

Research data are available on reasonable request.

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
