# Peer review of "Real-World Impact of SABR on Stage I Non-Small-Cell Lung Cancer Outcomes at a Scottish Cancer Centre"

_cancers, 2023, doi:10.3390/cancers15051431_

Round 1
Reviewer 1 Report
This is an interesting article demonstrating the realworld impact of SBRT on stage I NSCLC. There are some points to be further rephrased.
1. In the section of Method, the guideline or policy of SCAN lung cancer multidisciplinary meeting. And, following data need to be added.
- the proportion of pathological stage I cancer suspected with clinical stage II or more.
- the proportion of pathological stage II or more cancer suspected with clinical stage I cancer.
2. The differences between SABR and NRT reagarding to radiation dose and fraction as well as treatment aim are further described in the method section.
Author Response
We thank the reviewer for their positive feedback. Points have been addressed as below.
Comments and Suggestions for Authors
This is an interesting article demonstrating the real-world impact of SBRT on stage I NSCLC. There are some points to be further rephrased.
- In the section of Method, the guideline or policy of SCAN lung cancer multidisciplinary meeting.
Point taken. This has been included in the revised manuscript:
Line 106: All NHS Lothian patients discussed in the Southeast Scotland Cancer Network (SCAN) lung cancer multidisciplinary meeting between January 2012 and December 2019, diagnosed clinically with a stage I NSCLC were identified (35).
And, following data need to be added.
- the proportion of pathological stage I cancer suspected with clinical stage II or more.
- the proportion of pathological stage II or more cancer suspected with clinical stage I cancer.
Point taken. Analyses were undertaken on patients with a clinical diagnosis of stage I NSCLC on an intention to treat basis. This has been made clearer in the manuscript:
Line 107: “Patients with multiple synchronous or metachronous primary lung cancers were excluded. Patients upstaged at surgery were included in analyses on the basis of intention to treat as stage I NSCLC.”
and 145: “41 (9%) patients treated with surgery were upstaged.”
- The differences between SABR and NRT reagarding to radiation dose and fraction as well as treatment aim are further described in the method section.
Point taken. The manuscript has been amended:
Line 127: “Radical radiotherapy treatment status was defined as: CRRT - 55Gy in 20 fractions as fractionated dose; SABR – 54Gy in 3 fractions, 55GY in 5 fractions or 60Gy in 8 fractions. This is in keeping with previously reports (36)”
Reviewer 2 Report
The article of Stares et al. presents the impact of stereotactic body radiotherapy (SBRT) on survival in stage I lung cancer in a single Scottish institution from 2012 to 2019. The study is well-designed, includes a relatively large population, and the results are presented clearly. The authors conclude that the introduction of SABRT significantly improved survival in the entire study population, with the most significant impact in the surgical cohort.
Major comments
- The Authors attribute increased survival outcomes in particular periods exclusively to the introduction of SBRT. Although the impact of SBRT seems to be unquestionable, other potential factors warrant discussion (e.g. better perioperative care or progress in radiotherapy planning and delivery). Indeed, a major survival increase has been achieved in patients subjected to surgery, probably not only due to the enhanced selection of surgical patients. Of note, increased survival also icluded patients administered any radical radiotherapy (Fig. 2E).
- The authors state that the hospital database contains detailed clinical information for all patients with lung cancer within the analyzed period. Hence, it would be advisable to provide more data on the cohort of patients who did not receive radical treatment. How many of them received palliative treatment vs. best supportive care? This cohort is relatively large (around 32%), given that all series included only stage I patients. SABRT is the treatment of choice for elderly patients or those with poor PS or significant comorbidities. It seems that this opportunity has not been sufficiently exploited in the current series; the overall proportion of SABRT after full implementation of this method was only 18%, apparently lower than in other published series. Indeed, the introduction of SABRT has impacted the proportion of patients who did not receive radical therapy by merely 3% (33% vs. 30% in periods A and C, respectively). These figures need interpretation, as they may undermine the main conclusion of attributing increased overall survival only to the introduction of SABRT.
- Figure 1 may suggest a significantly higher efficacy of surgery over conventional radical radiotherapy and SBRT. The authors should clearly state that this difference is due to treatment selection.
Minor comments
- In the abstract, please provide median survival for all time periods (A, B, C) rather than only A and C.
- The authors use interchangeably two terms: “radical radiotherapy” and “conventional radical radiotherapy.” Whereas the second is clear, the first may be interpreted as either conventional radical radiotherapy or any radical radiotherapy, including SABR. To clarify, consider changing the abbreviation RRT to CRT for conventional radical radiotherapy.
- The authors state that approximately 20% of patients in Scotland present with stage I disease, but the respective reference (3) does not include stage distribution. This proportion seems somewhat high in an unselected country population (even though in the authors’ selected hospital series, this rate was even higher).
- Figure 3. In the legend: “any radiotherapy,” – you probably mean any radical radiotherapy.
- In Figure 4, Charlson comorbidity index >3, reverse A, B, and C periods.
- Supplemental Tables 1 and 2: explain in the legend time periods (A, B, C).
- Supplemental Table 2: explain abbreviations in the legend.
- Supplemental Figure 4: use in the legend full names rather than abbreviations
Author Response
The authors thank reviewer 2 for their time and very helpful feedback. Each point has been addressed in turn below. We welcome any further comments.
Comments and Suggestions for Authors
The article of Stares et al. presents the impact of stereotactic body radiotherapy (SBRT) on survival in stage I lung cancer in a single Scottish institution from 2012 to 2019. The study is well-designed, includes a relatively large population, and the results are presented clearly. The authors conclude that the introduction of SABRT significantly improved survival in the entire study population, with the most significant impact in the surgical cohort.
Major comments
- The Authors attribute increased survival outcomes in particular periods exclusively to the introduction of SBRT. Although the impact of SBRT seems to be unquestionable, other potential factors warrant discussion (e.g. better perioperative care or progress in radiotherapy planning and delivery). Indeed, a major survival increase has been achieved in patients subjected to surgery, probably not only due to the enhanced selection of surgical patients. Of note, increased survival also included patients administered any radical radiotherapy (Fig. 2E).
Point taken. The manuscript has been amended:
Line 273: “A potential confounder to these findings is the improvement in surgical techniques and perioperative care during the study time periods. However, the most significant reductions in surgical rates between time period A and C were seen in patients aged ≥85 years (5% v 0%), PS 2 (28% v 18%) and CCI ≥3 (24% v 7%). … Although survival improved between time periods A and C for patients treated with any radical radiotherapy, there was no significant change for patients treated with CRRT, suggesting this improvement was driven by treatment with SABR.”
- The authors state that the hospital database contains detailed clinical information for all patients with lung cancer within the analyzed period. Hence, it would be advisable to provide more data on the cohort of patients who did not receive radical treatment. How many of them received palliative treatment vs. best supportive care? This cohort is relatively large (around 32%), given that all series included only stage I patients.
SABRT is the treatment of choice for elderly patients or those with poor PS or significant comorbidities. It seems that this opportunity has not been sufficiently exploited in the current series; the overall proportion of SABRT after full implementation of this method was only 18%, apparently lower than in other published series. Indeed, the introduction of SABRT has impacted the proportion of patients who did not receive radical therapy by merely 3% (33% vs. 30% in periods A and C, respectively). These figures need interpretation, as they may undermine the main conclusion of attributing increased overall survival only to the introduction of SABRT.
Point taken. The manuscript has been amended:
Line 238: “We also note that between time period A and C rates of radical therapy increased by only 3% in the overall population and 6% in patients ≥75 years old. This is lower than that seen in a previous real-world observational study (24). Given that patients in the NRT cohort were older, less fit and more comorbid, we suspect that many had incidental lesions identified but weren’t fit for further investigation and management. Our institution has no thresholds for minimum lung function for SABR and, broadly, if a patient tolerates PET-CT they are likely to tolerate the delivery of SABR. Indeed, only 2% of patients in the NRT received any cancer direct palliative therapy, including high-dose palliative radiotherapy. This suggests that, in addition to the availability of new treatments, strategies to improve patient fitness or early detection of cancer are needed to improve radical treatment rates.”
- Figure 1 may suggest a significantly higher efficacy of surgery over conventional radical radiotherapy and SBRT. The authors should clearly state that this difference is due to treatment selection.
Point taken. The manuscript has been amended:
Line 249: “In our clinical practice surgery remains the treatment of choice for patients with stage I NSCLC. That patients treated with surgery in our cohort had significantly better survival than those treated with any radical radiotherapy likely reflects differences in treatment selection.”
Line 282: “Although survival improved between time periods A and C for patients treated with any radical radiotherapy, there was no significant change for patients treated with CRRT, suggesting this improvement was driven by treatment with SABR.”
Minor comments
- In the abstract, please provide median survival for all time periods (A, B, C) rather than only A and C.
The manuscript has been amended accordingly:
Line 38: “Median survival increased from 32.5 months in time period A to 38.8 months in period B to 48.8 months in time period C.”
- The authors use interchangeably two terms: “radical radiotherapy” and “conventional radical radiotherapy.” Whereas the second is clear, the first may be interpreted as either conventional radical radiotherapy or any radical radiotherapy, including SABR. To clarify, consider changing the abbreviation RRT to CRT for conventional radical radiotherapy.
Point taken. The manuscript has been amended throughout with an updated abbreviation of CRRT for conventional radical radiotherapy.
- The authors state that approximately 20% of patients in Scotland present with stage I disease, but the respective reference (3) does not include stage distribution. This proportion seems somewhat high in an unselected country population (even though in the authors’ selected hospital series, this rate was even higher).
Many thanks. The URL has been changed to the updated Public Health Scotland webpage.
- Figure 3. In the legend: “any radiotherapy,” – you probably mean any radical radiotherapy.
Point taken. The term “radical radiotherapy” has been replaced by “any radical radiotherapy throughout the manuscript.
- In Figure 4, Charlson comorbidity index >3, reverse A, B, and C periods.
Many thanks. The figure has been amended.
- Supplemental Tables 1 and 2: explain in the legend time periods (A, B, C).
This has been amended in the supplementary file
- Supplemental Table 2: explain abbreviations in the legend.
This has been amended in the supplementary file
- Supplemental Figure 4: use in the legend full names rather than abbreviations
This has been amended in the supplementary file
Reviewer 3 Report
The manuscript is well written and needs minor changes
1. How did the authors account for stage migration due to better imaging and patients who are screen detected vs who presented with symptoms
2.The figures need numbers at risk below the figures
Author Response
We thank the reviewer for their positive feedback. Comments are addressed below with reference to changes made to the manuscript.
Comments and Suggestions for Authors
The manuscript is well written and needs minor changes
- How did the authors account for stage migration due to better imaging and patients who are screen detected vs who presented with symptoms
We find no evidence of stage migration during the time period examined. This is described in the submitted manuscript (see line 145)
Screening for lung cancer has only recently been approved (September 2022) by the UK national screening committee. It is not currently performed in routine clinical practice in Scotland or the wider UK. None of the patients in this study would have been screen detected.
2.The figures need numbers at risk below the figures
Point taken. These have been added to the supplementary data.
Round 2
Reviewer 2 Report
The authors well addressed all comments raised in my review. My only minor suggestions is to explain in the Figure 2 legend the meaning of "Any radical radiotherapy"by adding "(CRRT or SABR).
Author Response
We are pleased that the reviewer has accepted our revised manuscript. The remaining query has been addressed as suggested. Many thanks on behalf of all the authors.